# Vast Gene Flow among the Spanish Populations of the Pest *Bactrocera oleae* (Diptera, Tephritidae), Phylogeography of a Metapopulation to Be Controlled and Its Mediterranean Genetic Context

**DOI:** 10.3390/insects13070642

**Published:** 2022-07-17

**Authors:** Esther Lantero, Beatriz Matallanas, M. Dolores Ochando, Carmen Callejas

**Affiliations:** 1Department of Genetics, Physiology and Microbiology, Faculty of Biological Sciences, University Complutense of Madrid, Av. José Antonio Novais 12, 28040 Madrid, Spain or esther.lanterobringas@ceu.es (E.L.); or beatriz.matallanas@universidadeuropea.es (B.M.); dochando@ucm.es (M.D.O.); 2Department of Basic Medical Sciences, School of Medicine, Universidad San Pablo-CEU, CEU Universities, Urbanización Montepríncipe, 28660 Boadilla del Monte, Spain; 3Department of Health Sciences, Faculty of Biomedical and Health Sciences, European University of Madrid, c/Tajo, s/n, 28670 Madrid, Spain

**Keywords:** *Bactrocera oleae*, cytochrome oxidase subunit I (*COI*), genetic diversity, haplotype, phylogeography, gene flow, dispersion

## Abstract

**Simple Summary:**

The output of olive industry at the Mediterranean Basin, headed by Spain, is huge worldwide. The olive fruit fly *Bactrocera oleae* is the major pest of olive orchards. The damages it causes become in considerable economic losses as well as a decrease in oil quantity and quality. A key question for the success of pest control strategies is the further knowledge about the species, and genetic data becomes essential for this purpose. The present work analyses more than 250 fruit flies from six different Mediterranean countries, showing relevant data about the genetic structure and gene flow of this damaging pest. These findings are helpful to improve the integrated pest management strategies according to the current European Guidelines.

**Abstract:**

Spain is the leading producer of olives and olive oil. Ninety-five percent of world production originate from Spain and other regions of the Mediterranean Basin. However, these olive-growing countries face a major problem, the harmful fly *Bactrocera oleae*, the main pest of olive crops. To improve its control, one of the challenges is the further knowledge of the species and populations dynamics in this area. A phylogeographic work is necessary to further characterise the levels and distribution patterns of genetic diversity of the Spanish populations and their genetic relationships with other Mediterranean populations. A 1151 bp fragment of the mitochondrial cytochrome oxidase subunit I (*COI*) gene has been analysed in over 250 specimens of the six main Mediterranean countries via sequencing. Genetic diversity parameters were high; 51 new haplotypes have been identified showing a geographical pattern across the Mediterranean area. The data revealed that olive fruit fly populations have been long time established in the Mediterranean Basin with two genetic groups. Gene flow seems to be the main process in shaping this genetic structure as well as fly’s colonisation routes that have paralleled those of the olive tree.

## 1. Introduction

The olive tree, *Olea europaea,* (Linnaeus 1753), is an iconic species of the Mediterranean Basin, where it covers more than nine million hectares of groves [1]. This cultivation makes up the backbone of the socio-economic and cultural life of many regions in this area, as well as an important source of healthy food. The olive oil is appreciated worldwide as a key component of the Mediterranean diet model, which has constituted a UNESCO Intangible Cultural Heritage since 2013 [2]. Concerning agricultural production, the economic relevance of the olive grove in Spain is unquestionable. It is the leading producer and exporter of table olives and olive oil, representing 45% of the world’s olive oil production [3]. 

Besides the economic magnitude, the olive cultivation also has social, territorial and environmental repercussions. Its environmental role is significant, mainly in terms of biodiversity, carbon sequestration, soil conservation and the generation of valuable agricultural landscapes at the current scenario of global change, which benefits pest species from increasing their growth rates [4]. The olive fruit fly (*olf*) *Bactrocera oleae* (*B. oleae*; Rossi 1790) is the major damaging pest of the *Olea* genus. Crop losses are due primarily to larvae feeding habits, which seriously reduces the fruits’ oil content and the organoleptic properties and causes the premature fruit drop. This tephritid is widespread throughout the Mediterranean Basin and the Middle East. It is also found in the South and East of Africa, India, and Pakistan; since the end of the 20th century, the olive fly has been also reported in California and Mexico and very recently (2019) in Hawaii [5]. 

The *B. oleae* populations need to be effectively controlled. The national and European competent institutions (Directives 2009/128/CE, RD 1311/2012 and BOE-A-2012-11605) request integrated control strategies that limit the use of phytosanitary products. For this challenge, a better understanding of the species is essential [6,7], with genetic questions such as genetic structure, population dynamics or gene flow being helpful for the development of control alternatives [8,9].

Earlier genetic studies on the *olf* populations yielded conflicting results on the genetic structure of Mediterranean *olf* populations. The debate over the number of genetic (sub)groups and the relationships among them was affected by the heterogeneity of sample sizes and the genetic markers used. Results vary from no evidence of genetic differentiation among populations (microsatellites and NADH mtDNA fragment [10]), a tendency towards differentiation (particularly Tunisian from European populations through RAPD markers [11,12] and intermicrosatellites [13]) to two or three genetic groups, depending on the survey (loci microsatellites [14,15,16]). All these studies were relevant to ascertain the genetic diversity patterns of this damaging pest species, but they also denoted that the structure of *B. oleae* is not yet fully characterised in the Mediterranean Basin, the hotspot of olive industry.

Notwithstanding, the characterisation of the Spanish *B. oleae* populations has been limited, since those surveys included few individuals from very restricted locations, despite the large area under olive cultivation in the country. Then, the first objective of this article was to increase the phylogeographic knowledge of this harmful pest in Spain and, secondly, its genetic relationships with other populations. Then, samples from other Mediterranean countries were included as external references to get a comprehensive scene of the genetic structure, demographic history and colonisation routes in this region. This survey has represented the most exhaustive study so far, with representative populations of the entire Spanish geography using the cytochrome oxidase subunit I (*COI*) mitochondrial marker. This gene is the universal barcode for animal species identification, but its utility goes far beyond taxonomy, given the high informative genetic variation it holds at the intraspecific level [17].

## 2. Materials and Methods

### 2.1. Sampling

Olive flies were sampled from 15 Spanish populations across the whole productive area (Table 1 and Figure 1). When possible, infested olives were collected, allowing larvae development in climatic cameras at the laboratory. The emergent adults were then stored at −20 °C. Additional samples from Portugal, Greece, Israel, Italy and Tunisia were also analysed, and adults were picked by different collaborators and directly sent in ethanol (70%) and conserved at 4 °C until the DNA extraction. 

### 2.2. Methods

#### 2.2.1. DNA Isolation, Amplification and Sequencing

Individual genomic DNA was extracted using the DNeasy Blood and Tissue Kit (Qiagen Valencia, Los Ángeles, CA, USA) following the instructions given by the manufacturer with minor modifications. The concentration and purity of the DNA were estimated both electrophoretically and by absorbance at 260–280 nm with a NanoDrop ND-100 (NanoDrop Technologies, Wilmington, NC, USA). 

A 1.75 Kbp fragment of the mitochondrial gene *COI* was amplified with the C/N 2769 and LCO 1490 primers at a final volume of 12.5 µL [18,19]. PCR reactions included 10 ng of genomic DNA, 1× buffer, 4 mM MgSO_4_, 200 µM of each dNTP, 0.65 µM of each primer and 1 U of the high-fidelity Vent^®^ DNA polymerase (New England Biolabs, MA, USA). The PCR program included an initial denaturation step at 94 °C for 5 min, followed by 35 cycles of 94 °C for 45 s, 60 °C for 1 min and 72 °C for 1 min, plus a final extension step of 72 °C for 10 min. All reactions were performed in a LabCycler apparatus (SensoQuest, Göttingen, Germany).

The amplicons were firstly checked on 0.8% agarose gels with TBE buffer (40 mM Tris-Borate, 1 mM EDTA, pH 8.0), stained with ethidium bromide (0.5 mg/mL). Then, they were purified in a single-step enzymatic clean-up method with enzymes exonuclease I (Exo I) and FastAP^TM^ (thermosensitive alkaline phosphatase) (Thermo Fisher Scientific, MA, USA). Both strands were sequenced using the BIG Dye^®^ Terminator Cycle Sequencing Ready Reaction Kit (Applied Biosystems, Inc., USA) in a 3730 DNA Analyser (Applied Biosystems, Inc., USA) at the Genomics Unit of the University Complutense of Madrid, with the same primers employed for amplification. 

#### 2.2.2. Data Analyses

The *COI* sequences were aligned with CLUSTAL W software and edited with BioEdit v.7.0.9.0 [20,21]. To control for possible nuclear mitochondrial pseudogenes (numts) and double peaks, all chromatograms were carefully checked. Sequences were also theoretically translated into amino acids searching for unexpected stop codons and indels, as recommended [22,23]. After alignment and edition, *COI* sequences were trimmed to a final length of 1151 bp and deposited in GenBank under accession numbers of KC005742-KC005762, KP704354-KP704404 and JX073648.

The parameters of the haplotype diversity (*Hd*), the nucleotide diversity (π) and the number of segregating sites (S) were calculated with DNAsp v 5.10.01 program [24] and the haplotype frequencies for each population were analysed and recorded on a map. The reticulated graphs are considered the most suitable method to establish connections among closely related sequences [25]. Then, median-joining (MJ) networks were used to reconstruct genetic relationships using Network software v.10 [26]. Likewise, to get a broader view of the genetic structure of *B. oleae* populations, the network was repeated discarding the exclusive haplotypes and incorporating *olf COI* sequences from GenBank with accession codes GU108459-GU108479 and HQ677155-HQ677162 and *Bactrocera dorsalis* (*B. dorsalis*) DQ845759 as an outgroup [27,28,29].

To infer the genetic differentiation between populations, F_ST_ fixation indices were calculated with Arlequin v 3.5.12, and statistically significant levels were determined applying a sequential Bonferroni correction [30,31,32]. These pairwise F_ST_ fixation indices were also used in a principal coordinate analysis (PCoA) [33], employing NTSYSpc v2.10q software for visualising the genetic structure of populations [34]. Gene flow (Nm) among populations was estimated with the approximation *F_ST_* ≈ 1/(1 + 2Nm) [35]. 

The analysis of molecular variance (AMOVA) was performed to check the hierarchy of the genetic structure detected at the Mediterranean Basin, using Arlequin software with 10,000 permutations. Additionally, to explore patterns of genetic divergence in more detail, the spatial analysis of molecular variance (SAMOVA) was also applied defining homogeneous groups and maximising the proportion of genetic variance due to differences among groups of populations [36]. These analyses were conducted with arbitrary partitions of the populations into different groups (k = 2–6).

Finally, Fu’s Fs, R_2_ and Tajima’s D neutrality tests were performed with the genetic groups revealed by PCoA and SAMOVA [37,38,39]. Significance values were calculated by coalescent computer simulations (10,000 replicates), using a modification of the tree routine as provided by the DnaSP v5.10 software [24,40]. Mismatch distributions analyses were also estimated to check expansion events [41]. The sum of squared deviation (SSD) was calculated to assess whether the distribution fitted the sudden expansion model. A thousand bootstrap replicates were used to generate an expected distribution using this model with Arlequin v 3.5.12 [31]. Evolutionary time, τ, is the time in generations elapsed since the last expansion. It was determined using the equation τ = 2*u*t where τ is estimated via mismatch distribution analysis and expressed in units of evolutionary time, *t* is the time in generations, and *u* = µk, in which µ is the substitution rate in the specific DNA region, and k the length of the analysed sequence. A generation time of 0.25 years and a substitution rate of 2.3% of divergence per million years for the *COI* gene were applied [27,42].

## 3. Results

The study of 15 samples representative of the distribution range of *B. oleae* in Spain has revealed a high diversity at the mitochondrial level. The 1151 bp sequence analysis detected 45 polymorphic sites that defined 49 haplotypes, of which 34 were exclusive, i.e., they were present in a single fly. These haplotypes were separated by 1–3 SNPs (Single Nucleotide Polymorphism), being the most frequent H1 followed by H2 and H15. In the Spanish populations as a whole, the haplotype diversity (*Hd*) resulted 0.864, and the nucleotide diversity (*π*) was 0.00151 (Table 2). Focusing on populations, *Hd* values ranged from 0.955 to 0.755 (SPA6−SPA12), and *π* varied from 0.00176 to 0.00112 (SPA4−SPA7). The SPA5 sample showed very low values in relation to the range found (*Hd* = 0.644 and π = 0.00089). 

When the samples from other five Mediterranean countries were also analysed, up to 60 SNPs were recorded (81.66% transitions), of which 26 were parsimony informative. Although 73 *COI* haplotypes were found, differing at 1–9 mutational steps, six encompassed 63.63% of the analysed flies, with not only H1, H2, H17 and H15 being the most common as in Spain, but also H21 and H22. The remaining were at very low frequencies, and some were population-specific. Overall, *B. oleae* showed a *Hd* value of 0.908 and a *π* value of 0.00259 (Table 2), although the highest haplotype and nucleotide diversities were found in the Iberian Peninsula populations. 

This haplotype diversity was represented on a map with pie charts on the coordinates of each population sampled (Figure 2). The haplotypic distribution was not homogeneous and reflected a marked geographical pattern. H1 (24%), H2 (7.7%) and H15 (6.2%) were characteristic of the westernmost *olf* populations from Spain, Portugal and Tunisia (TUN1 and TUN3) and also Greece (GRE1 and GRE2). H22 (6.2%) was detected in all eastern *olf* populations, and H21 (6.5%) was detected in Israel, Greece (GRE1) and two populations from Tunisia (TUN1 and TUN2). The most widely distributed haplotype was H17 (13%), found in some Spanish populations and in all ones from Tunisia, Italy and Greece.

Regarding the genetic structure of this pest in Spain, the *F_ST_* estimates were practically null when comparing the 15 *olf* samples, with 0.101 being the highest one (Appendix A). These low *F_ST_* values are directly related to an unrestricted gene flow among the *olf* Spanish populations. The same occurred, when samples of Spain and Portugal were contrasted. The AMOVA analysis confirmed this outcome (Table 3). Spanish and Portuguese *B. oleae* populations constituted a single (meta)population. When comparing the Spanish populations with the others (Appendix A), low *F_ST_* estimates were found in general among Spain, Portugal, Tunisia and Italy (0–0.210). 

The PCoA provided a two-dimensional graphical view of the relationships among these populations (Figure 3). The X-axis, accounting for the 97% of this genetic variation, distinguished two main genetic clusters. The first was made up of all the Spanish and Portuguese samples, which are very close to each other, and included the Tunisian, Italian and Greek (1 and 2) populations. The second cluster comprised the Israel and Greece 3 samples. Some substructures are shown in the former genetic cluster, with the Spanish and Portuguese samples differing from Tunisian and Greek (1 and 2) ones. The Italian sample occupied an intermediate position along the X-axis between SPA/POR and GRE/TUN subgroups, and it was mainly discriminated along the Y-axis. Likewise, the SAMOVA confirmed this structure identifying two as the optimal number of genetic clusters (Appendix A). The AMOVA performed considering such two main genetic clusters, group I and group II, rendered 63% of variation among groups (*p* < 0.0001; Table 3). The estimated average gene flow between the populations of both haplogroups was minimum (Nm: 0.27 (*F_ST_* = 0.649)), whereas the gene flows among population within the two groups Nm were 3.14 and 3.51, respectively. 

The MJ networks exhibited a star-shape topology with the three haplotypes (H1, H17 and H22), from which the other ones were radiated and detected (Appendix A). From the H1 haplotype derived 18 exclusive haplotypes of Spanish, Portuguese and Italian populations. The second most frequent haplotype, H17, differing from H1 at one SNP, had the widest geographical distribution, and it was designated as the ancestral haplotype. Haplotypes H1 and H17, together with the 57 haplotypes derived, fell under group I. By contrast, H22, as one of the most frequent haplotypes, along with H21, both of which were separated by two SNPs, at low frequency and exclusively derived, genetically composed group II. 

For a more complete overview of the genetic structure of the olive fly pest, a second haplotype network was elaborated by including 31 additional *COI* sequences and removing exclusive haplotypes. The network (Figure 4) accorded with the pattern of geographical distribution of the *COI* genetic diversity detected above. Notwithstanding, there were some interesting aspects to be noted. The additional Italian (H1 and H17) and Greek (H2 and H17) *COI* sequences, besides the Algerian (H2) and Moroccan ones (H74), joined the Mediterranean group I. The sequences from Turkish and Israeli flies (H21, H22, H23, H52 and H54) were integrated in haplogroup II. The USA flies also carried the haplotype H22, typical of this haplogroup. The haplotype H76 was characteristic of flies from Kenya, South Africa and Turkey; it was placed at one SNP apart from the Mediterranean and the widely distributed H17. Finally, a different haplogroup comprised the *COI* sequences from Pakistani flies with haplotypes H77, H78 and H79, separated by four or five polymorphisms from the Middle East and African haplotype H76. *B. dorsalis* rooted this network, allowing designating the Pakistani haplotypes as ancestral. It also confirmed that haplotype 17 was the ancestral of the 73 detected in the Mediterranean populations analysed.

The Spanish olive flies experienced a demographic expansion that fitted a sudden expansion model (SSD = 0.0022, *p* > 0.05; Appendix A), according to the unimodal mismatch distribution and the inferences from negative D values of D Tajima, Fu’s *Fs* and very low R2, which were statistically significant or close to significance. This expansion might occur about 66,000 years ago. Likewise, haplogroup I might have a demographic expansion under the sudden expansion model (SSD = 0.0027; *p* > 0.05) occurring about 49,000 years ago. Haplogroup II was adjusted to a spatial expansion model assuming a constant population size (SSD = 0.0122; *p* > 0.05) that might take place about 61,500 years ago. When considering Mediterranean *B. oleae* samples as a single population, it fitted to a spatial expansion model (SSD = 0.0048; *p* > 0.392), estimated about 61,000 years ago. In short, these results pointed to an expansion of the populations of olive fly populations in the Mediterranean dated at the Pleistocene.

## 4. Discussion

The study of 15 samples from the entire Spanish geography showed the mitochondrial diversity to be high in *B. oleae* (Table 2), a pest causing havoc in the olive production. From the 1151 bp nucleotides analysed, 45 were variable yielding 49 haplotypes of which H1, H2 and H15 were the most frequent and more widely distributed throughout Spain (Appendix A). The haplotypic and nucleotide diversities ranged from 0.864 to 0.91 and 0.00151 0.91 and 0.0026, respectively, when the remaining populations were also considered. Such values are similar or slightly higher than those estimated in other works of Iberian samples with mitochondrial markers (*Hd*: 0.88 [43], 0.79 ± 0.4 [10] and 0.82 [44]). 

Genetic diversity estimates from the *COI* marker in some polyphagous Dacini revealed a haplotype (h) and a nucleotide (π) diversity of 0.92075 ± 0.017 and 0.01083 ± 0.00052 in the fruit pest *Bactrocera zonata* from seven Iranian locations [45]. For *Bactrocera correcta,* parameters were h = 0.9337 and π = 0.0132 from 15 Thai populations and around 0.95 and 0.008 in a compilation of *B. dorsalis* surveys, both of which were economically important polyphagous fruits pests in Asia [46,47]. Likewise, in Asian populations of the *Zeugodacus. Tau* complex, h and π estimates were 0.8954 and 0.0033, respectively [48]. However, concerning *B. oleae*, of which the larvae are totally dependent on the olive fruit, despite its reduced environmental diversity, the results reflected a high genetic variability, in the similar range as the polyphagous dacines ones. Our results suggested that *B. oleae* does not conform the expectations of low genetic diversity given its limited environmental diversity [49]. This outcome could be a characteristic of the species.

The considerable neutral genetic variability observed in the Spanish *olf* populations, where most SNPs were silent transitions, and the presence of exclusive haplotypes may be due to large population sizes. This hypothesis is supported by the high fecundity of the females, which lay more than 200 eggs, with two to five generations per year [50]. This fact is favoured by the current Mediterranean temperature and humidity conditions, as well as by the large areas of olive cultivation. Another factor relies on the long-established status of the species in this area. To illustrate, charcoal from cultivated specimens of early Neolithic age showed that the olive tree is the earliest cultivated temperate fruit in Eastern Spain, where the wild olive tree has been indigenous since last glaciation and *olf* have already parasitised the wild variety [51,52]. 

Iberian Peninsula populations were not differentiated (*F_ST_*, PCoA, AMOVA and SAMOVA analyses), regardless of other variables such as the olive grove management regimes. Therefore, the Spanish and Portuguese *B. oleae* populations constituted a single genetic meta-population on a country-wide geographical scale, reliably supporting our previous surveys [12,44]. These results are relevant for pest management, since the Spanish olive extension covers more than 2.5 million hectares, which could facilitate gene flow and local *olf* dynamics of decline/recolonisation. Likewise, the adaptive history of the species to long and current local conditions cannot be avoided to improve an efficient pest control [53].

Genetic analyses revealed the Spanish olive fly samples to be genetically very similar not only to Portuguese ones, but also to samples from Italy, Tunisia, particularly TUN3, and part of Greece (Table 3 and Appendix A; Figure 2, Figure 3, Figure 4 and Appendix A). All of them conformed the genetic group I. The *olf* populations from Israel and GRE3 constituted the genetic group II and the populations of both clusters were weakly interconnected (Nm: 0.27; *F_ST_* = 0.649). In turn, there was additional regional sub-structure discriminating the populations of Spain and Portugal from those of Tunisia and Italy, which constituted genetically distinct entities, according to our group’s preliminary work [44]. It was noteworthy that the Northern Greek populations lumped into this cluster because of their genetic similarity to Tunisian ones, whereas the southernmost sample (GRE3) had a genetic pattern more like the Israeli ones. Consistent with this finding, the inclusion of an additional Greek *olf* sample from Thessaloniki in the analyses (n = 8) revealed its integration into group I [28]. None of the previous studies involving large Greek *olf* samples showed genetic differentiation [14,16]. These outcomes illustrated the sensitivity of the *COI* gene fragment used. The employment of a longer fragment than those usually used, located at the 5’ region of the gene, which is more variable, has demonstrated the suitability of this marker for inferring new phylogeographical features of this damaging pest. Traditionally in dacines and mainly on pest species, much works has utilised universal *COI* primers for barcoding and genetic diversity assessment [19,54,55] and references therein.

Gene flow is the main factor that has shaped the observed genetic structure overall. The intensive cultivation of olive trees in the Mediterranean Basin, accounting for 90% of the world’s olive oil production, favours these genetic diversity patterns. Olive flies have been reported to be able to cover considerable distances, depending on fruit availability; therefore, in cases of shortage of olives, adults may travel several kilometers in a few days [56,57,58,59]. The human-mediated dispersal should also be considered. Although most trade occurs with olive oil, olives from different localities are used in olive mills where usually the olives (some infested) remain for a few days before maceration and extraction [11]. Time is enough for the adults to emerge and infest the surrounding orchards, modifying the genetic composition of local populations, as Skouras et al. described at populations from Greece, Cyprus and Crete [57].

The additional haplotype network included 38 sequences from Genbank, in an effort to integrate and compare the results from all studies concurrently. This network corroborated the patterns of distribution of genetic diversity we detected in the Mediterranean, but also enhanced it (Figure 4). The additional Italian (H1 and H17) *COI* sequences joined the Mediterranean group I. Greek (H2 and H17) haplotypes also fell into this genetic group, as well as the Algerian (H2) and Moroccan (H74) ones. The sequences from Turkish and Israeli flies (H21, H22, H23, H52 and H54) were integrated in haplogroup II, but also the USA flies, which held H22. This outcome confirmed the recent American infestation from the Eastern Mediterranean area, as suggested using nuclear markers, although not enough time has elapsed to differentiate these olive flies from the Mediterranean ones [12,16]. 

The Mediterranean and widely distributed H17 are the ancestral of the 73 detected in the Mediterranean populations we have analysed (Appendix A and Figure 4). However, it is worth to highlight the haplotype H76, placing one SNP apart. H76 is characteristic of flies from Kenya, South Africa and Turkey. This haplotype occupied a basal position in the network rooted by *B. dorsalis*, a phylogenetically very close tephritid species. Then, H76 is the one of the ancestral *olf COI* haplotypes until now reported. 

Lastly, the haplogroup of the Pakistan samples (H77, H78 and H79) was clearly apart, differing by up to five SNPs from the Middle East and African *olf* haplotypes, which seemed to indicate an early divergence, as suggested [10,16,27]. Indeed, it is considered as *B. oleae* var. *asiatica.* Thus, in this Asian easternmost group, an early isolation, small population size and the genetic drift shaped the current structure of this population. Notwithstanding, the small sample size stressed the need to analyse a larger one for further inferences.

Therefore, the colonisation routes followed by *olf* populations is another relevant factor responsible for the distribution of genetic diversity revealed by the mitochondrial marker. Some authors suggested a westward expansion of the pest, in parallel with the expansion of olive cultivation, following the trade routes of the Phoenicians, Greeks and Romans [14,16]. In contrast, other authors claimed that the *olf* already parasitised the wild variety. Then, the genetic structure of *B. oleae* populations detected had a more ancient component. The olive tree species complex originated in Central−Eastern Africa and diverged into two subspecies, *cuspidata*, typical from South Africa and Asia and *europaea*, the Mediterranean subspecies around 4.5 million years ago in the Pliocene [60,61,62]. Although the geographical origin of *B. oleae* is still controversial, this species has probably arisen in sub-Saharan Africa. The genetic results seemed to indicate that the fly’s colonisation routes paralleled those of the olive tree as genetically distinct sub-Saharan, Mediterranean and Pakistani populations have been identified (Figure 4), supporting the suggestion elsewhere [27,63]. Likewise, the closest species to *B. oleae* are also of African origin [64,65]. Therefore, Central−East Africa seem to be the origin of this species infesting olive trees.

The olive wild trees of *Olea europaea* (*O. e.*) *subsp. europaea* diverged between 284,000 and 139,000 years ago, before the last glaciation, as reflects its genetic structure, according to the glacial refugia of the three Mediterranean peninsulas. In this scenario, demographic inferences (Appendix A) have pointed an expansion of *B. oleae* populations in Spain to occur in the Pleistocene around 66,000 years ago, when climatic fluctuations had a major impact on the distribution and evolution of species in temperate regions [66]. The same trend was seen in the Mediterranean genetic groups I and II detected, although the last estimated expansion in the Eastern group was slightly earlier, in agreement with a previous result suggesting the *olf* to be in Mediterranean olive trees for at least 400,000 years. As the wild olive trees were gradually replaced by cultivated olive ones, the *olf* moved to host domesticated olive groves, where it found a better habitat.

In short, the mitochondrial analysis has been extremely useful in the phylogeographic study of the *olf*, revealing very high values of genetic diversity and unrestricted gene flow in Spanish populations. A marked pattern of geographic distribution in the Mediterranean Basin and two major genetic groups have been identified, with a patent substructure, mainly due to gene flow and colonisation routes. These genetic results seemed to indicate that the fly’s colonisation routes paralleled those of the olive tree, with *olf* populations being long established in the Mediterranean Basin. The use of this *COI* genetic marker provides robustness to the results obtained, with a matrilineal inheritance and lack of recombination that provides a more ancient view of the evolutionary process occurring, allowing a sharper knowledge about the *olf* genetic structure in this area. All the information obtained in this study may be relevant in the major challenges that control programs of *B. oleae* have to face in the scenario of global trade and global change.

## 5. Conclusions

The populations of *B. oleae* analysed displayed high values of genetic diversity and unrestricted gene flow in Spain. The data revealed *olf* populations have been long time established in the Mediterranean Basin. A marked pattern of geographic distribution in the Mediterranean basin and two major genetic groups has been identified with two genetic groups. Gene flow seems to be the main process in shaping this genetic structure as well as fly’s colonisation routes that have paralleled those of the olive tree. These findings are helpful to improve the integrated pest management strategies according to the current European Guidelines.

## Figures and Tables

**Figure 1 insects-13-00642-f001:**
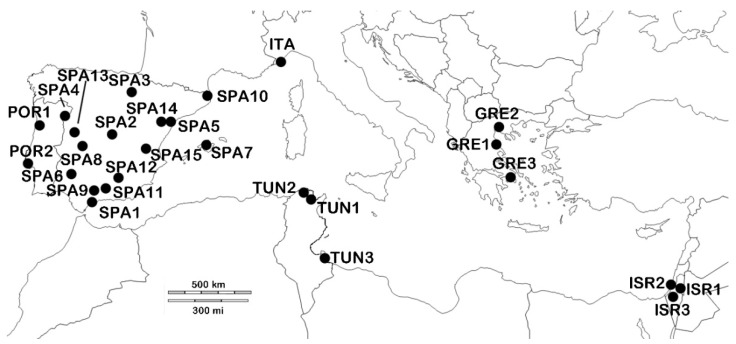
Map showing the *Bactrocera oleae* (*B. oleae*) populations sampled.

**Figure 2 insects-13-00642-f002:**
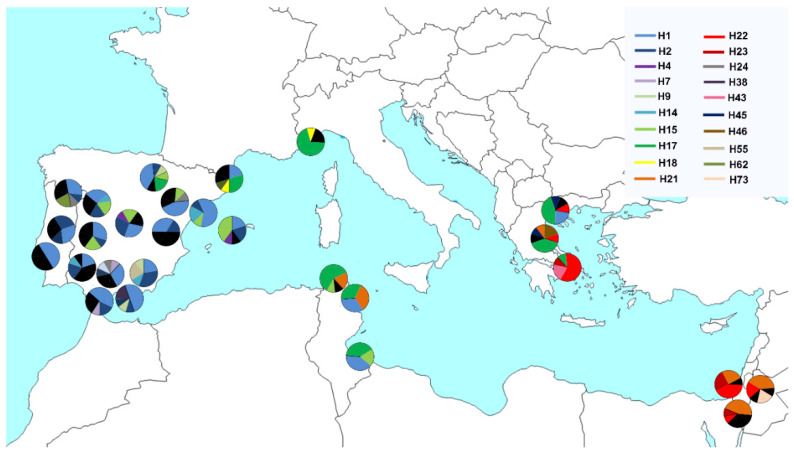
Pie graphs showing the distribution and relative proportion of *COI* haplotypes for each olive fly population sampled. Colours represent the haplotypes found. Exclusive haplotypes were denoted in black colour.

**Figure 3 insects-13-00642-f003:**
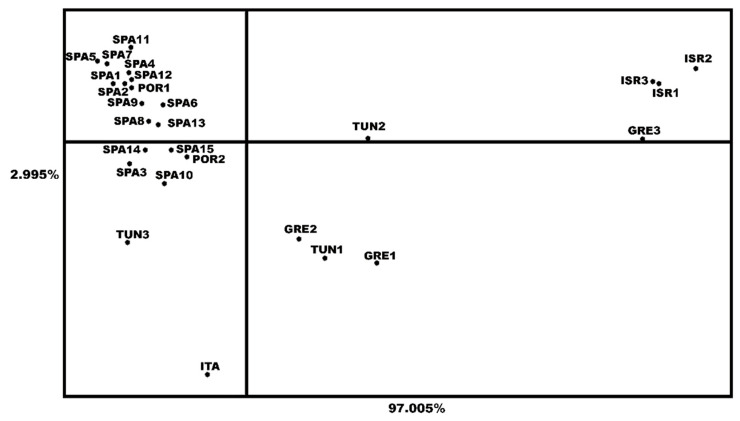
Principal co-ordinates analysis, based on the *F_ST_* indices, of the 15 Spanish *B. oleae* populations analysed plus 12 from other olive oil producer countries. The Eigenvalues for each principal component are listed besides each axis.

**Figure 4 insects-13-00642-f004:**
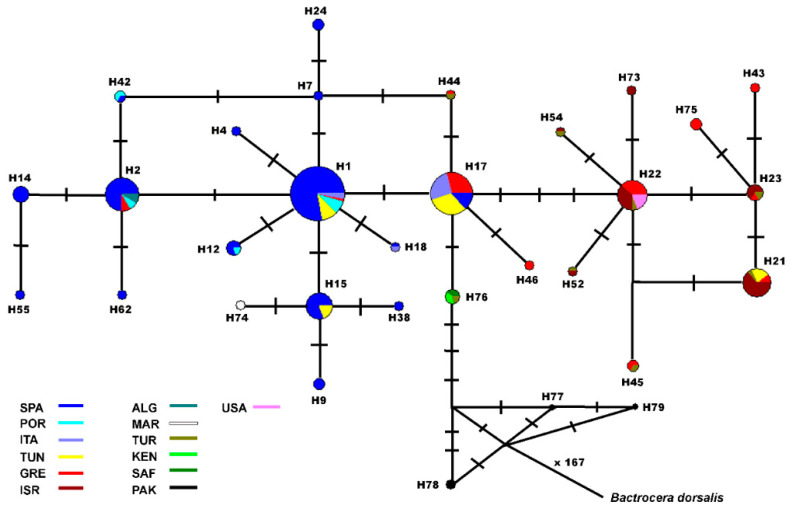
Haplotype network of the concatenated *COI* haplotypes generated by the “median-joining” method. The area of the circles is proportional to the haplotype frequency. Bars correspond to mutational steps between haplotypes. Colours represent the sampled countries. ALG, Algeria; MOR, Morocco; KEN, Kenya; SAF, South Africa; PAK, Pakistan.

**Table 1 insects-13-00642-t001:** Populations of *B. oleae* analysed with the code assigned, locality, geographical coordinates (LAT, latitude; LONG, longitude) and sample size (N). SPA, Spain; POR, Portugal; ITA, Italy; TUN, Tunisia; GRE, Greece; ISR, Israel.

CODE	Locality, Country	LAT	LONG	N
SPA1	Ronda, Málaga, ES	36.6587	−4.7603	11
SPA2	Morata de Tajuña, Madrid, ES	40.2275	−3.4369	11
SPA3	Arróniz, Navarra, ES	42.4222	−2.0913	10
SPA4	Aldeadávila de la Ribera, Salamanca, ES	41.2183	−6.62	10
SPA5	Tortosa, Tarragona, ES	40.811	0.5209	10
SPA6	Montemolín, Badajoz, ES	38.1552	−6.2069	10
SPA7	Mallorca, Islas Baleares, ES	39.6952	3.0175	10
SPA8	Castañar de Ibor, Cáceres, ES	39.6277	−5.4166	10
SPA9	Campus de Rabanales, Córdoba, ES	37.2647	−4.6327	10
SPA10	El Cortalet, Gerona, ES	42.2253	3.0970	10
SPA11	Íllora, Granada, ES	37.3461	−3.8727	9
SPA12	La Iruela, Jaén, ES	37.9469	−2.9583	10
SPA13	Lagunilla, Salamanca, ES	40.3246	−5.9687	10
SPA14	La Portellada, Teruel, ES	40.89	−0.0336	9
SPA15	Requena, Valencia, ES	39.4878	−1.1003	6
POR1	Fundao, PT	40.1369	−7.4994	7
POR2	Lisboa, PT	38.7069	−9.1356	6
ITA	Diana Marina, Liguria, IT	43.9098	8.0818	10
TUN1	Sidi Thabet, TN	36.9081	10.0222	10
TUN2	Tunisia, TN	36.7916	10.0634	6
TUN3	Zarzis, TN	33.523	11.0852	10
GRE1	Agia, GR	39.7188	22.7550	10
GRE2	Tesalonica, GR	40.6393	22.9446	9
GRE3	Atenas, GR	37.9791	23.7166	10
ISR1	Jerusalem, IL	31.7383	35.2137	10
ISR2	Rehovot, IL	31.8927	34.8112	12
ISR3	Lahav Forest, IL	31.3725	34.8408	11

**Table 2 insects-13-00642-t002:** Genetic diversity parameters of mtDNA cytochrome oxidase subunit I (*COI*) from the *B. oleae* populations analysed. *S*, the number of polymorphic sites; *h*, the number of haplotypes found; *Hd*, haplotype diversity; *π*, nucleotide diversity. SPA, Spain; POR, Portugal; ITA, Italy; TUN, Tunisia; GRE, Greece; ISR, Israel; SPAIN, the 15 Spanish samples together.

POP	*s*	*h*	*Hd*	*π*
SPA1	6	6	0.800	0.00117
SPA2	6	6	0.872	0.00130
SPA3	5	6	0.844	0.00114
SPA4	8	6	0.888	0.00176
SPA5	3	4	0.644	0.00089
SPA6	9	8	0.955	0.00209
SPA7	4	5	0.822	0.00112
SPA8	7	7	0.911	0.00158
SPA9	9	8	0.933	0.00170
SPA10	8	7	0.911	0.00162
SPA11	5	5	0.861	0.00164
SPA12	6	5	0.755	0.00154
SPA13	6	7	0.911	0.00151
SPA14	8	6	0.833	0.00154
SPA15	6	5	0.933	0.00174
SPAIN	45	49	0.864	0.00151
POR1	5	5	0.904	0.00157
POR2	4	2	1.000	0.00348
ITA	5	4	0.533	0.00114
TUN1	8	4	0.644	0.00207
TUN2	6	3	0.800	0.00278
TUN3	2	3	0.711	0.00077
GRE1	8	6	0.844	0.00236
GRE2	7	5	0.805	0.00193
GRE3	5	4	0.644	0.00124
ISR1	5	5	0.822	0.00160
ISR2	3	4	0.757	0.00096
ISR3	7	7	0.818	0.00171
SPECIES	60	73	0.908	0.00259

**Table 3 insects-13-00642-t003:** Analysis of molecular variance (AMOVA). The molecular variance was estimated among populations, among populations within groups and within populations.

Source of Variation	Variance Components	Percentage of Variation	*p*-Value
Spanish and Portuguese populations (Iberian Peninsula)			
**Among populations**	0.01160	1.31	>0.0500
**Within populations**	0.87064	98.69	
Mediterranean populations			
**Among populations**	0.625	41.38	<0.0001
**Within populations**	0.886	58.62	
2 genetic groups: group I: Iberian Peninsula, Italy, Tunisia and Greece 1/2; and group II: Greece 3 and Israel			
**Among groups**	1.707	62.26	<0.001
**Among populations**	0.133	4.87	<0.001
**Within populations**	0.901	32.87	<0.001

## Data Availability

*COI* DNA sequences were deposited on GenBank data bases under the accession numbers KC005742-KC005762, KP704354-KP704404 and JX073648.

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
