# Peer review of "Vast Gene Flow among the Spanish Populations of the Pest *Bactrocera oleae* (Diptera, Tephritidae), Phylogeography of a Metapopulation to Be Controlled and Its Mediterranean Genetic Context"

_insects, 2022, doi:10.3390/insects13070642_

Round 1

Reviewer 1 Report

Lantero et al. Insects

The authors have studied a 1151 bp fragment of the COI gene in 250 individuals belonging to the insect species Bactrocera olae in six Mediterranean countries. This species is a major pest of olive trees. They found a high molecular diversity in the populations analyzed. According to the haplotypes obtained, they also found two main groups of populations, where one of them is composed of a couple of subgroups. The authors deduced that the main process shaping the genetic structure observed is gene flow. Finally, they concluded that Mediterranean Bactrocera olae populations were established long time ago.

The manuscript is properly written, data are very valuable and the Discussion is general and interesting for potential readers of Insects journal.

Although the manuscript is interesting and I like it, I consider that minor corrections are needed.

- The name of genes has to be in italics. So, it is needed to correct “COI” in the whole manuscript.

- Line 39. Delete one “The”.

- Line 51. A link is necessary between both sentences.

- Lines 54-55. Is the species found always in olive tree?

- Line 70. I think that “2008” has to be deleted.

- Line 91. Samples from Portugal, Greece, Italy and Tunisia were collected by the authors? Please, clarify this point.

- Table 1. “Locality, Country” has to be centered.

- Table 1. “Baleares Isles” is not correct. Please, write the name in English (Balearic Islands) or in Spanish (Islas Baleares) or in Catalan (Illes Balears) language.

- Line 162. I think it has to be: “evolutionary time, t is the time in generations”.

- Line 167. This sentence has to be deleted.

- Lines 174-175. I think it is much better to eliminate de blank space between SPA and the number. Thus, it will be consistent with the abbreviations in Table 2.

- Table 2. SPA9. Add one “0” at the end of 0.0017.

- Table 2. The meaning of Spain in bold is not clear. Authors have to explain that this is whole sample from this country.

- Line 202. I figure out that the word “flow” is needed after “gene”.

- Table 3. The groups are confusing. Why are there two group 1? Spanish samples are included in the group 1 “Mediterranean populations” or not?

- Line 216-217. In my opinion, the position of Italian sample is not so clear as explained in the text. Please, clarify this point in the sentence. 

- Line 244. A “.” is needed after “haplogroup II”.

- Line 306. Delete one blank space after “decline”.

- Line 342. Add a “,” after Mediterranean.

-  Although gene flow is important for the genetic structure of the analyzed populations, I miss some discussion on the potential role of natural selection maintaining the differences between the observed groups.

- References section. There are small mistakes. It has to be revised.

Reviewer 2 Report

The minimum number of individuals required for a population genetics study is 20 per a population. There may be problems in group interpretation due to sampling bias, especially in AMOVA..

Lines 33 and 408:

“Gene flow seems to be the main process in shaping this genetic 33 structure as well as fly's colonization routes that would have paralleled those of the olive tree.” -> This sentence is awkward because gene flow generally makes regional populations closer and eventually makes them similar like a population. In your study, Italian, Greek or Turkish group is largely different from Spanish group. I think there are three or more genetic partitions in your PCA result. Can you define that there is a gene flow between them all? The region where the gene flow appeared seems to be valid only for the population of the Iberian Peninsula.

How about ‘unrestricted or restricted gene flow…’?

In title: Bactrocera oleae -> be italicized

Figure 4: An explanation of what haplotype is optionally presented is essential to the legend. Why were other haplotypes excluded?

Lines 200 and 202: Fst -> be italicized

Figure 3 legend. Are those Eigenvalues? I wonder how x- and y-axis can account for 100%? This is weird.

-end-
